# Antioxidant Mechanisms of the Protective Action of Selenase in Experimental Chronic Generalized Periodontitis

**DOI:** 10.3390/cimb47030186

**Published:** 2025-03-12

**Authors:** Valeriy Salnykov, Igor Belenichev, Lyudmyla Makyeyeva, Dmytro Skoryna, Valentyn Oksenych, Oleksandr Kamyshnyi

**Affiliations:** 1Department of Surgical and Propaedeutic Dentistry, Zaporizhzhia State Medical and Pharmaceutical University, 69035 Zaporizhzhia, Ukraine; salnikov.v.i@zsmu.edu.ua; 2Department of Pharmacology and Medical Formulation with Course of Normal Physiology, Zaporizhzhia State Medical and Pharmaceutical University, 69035 Zaporizhzhia, Ukraine; 3Department of Histology, Cytology and Embryology, State Medical and Pharmaceutical University, 69035 Zaporizhzhia, Ukraine; makyeyeva.l.v@zsmu.edu.ua; 4Department of Pharmaceutical, Organic, and Bioorganic Chemistry, Zaporizhzhia State Medical and Pharmaceutical University, 69035 Zaporizhzhia, Ukraine; 5Broegelmann Research Laboratory, Department of Clinical Science, University of Bergen, 5021 Bergen, Norway; 6Department of Microbiology, Virology, and Immunology, I. Horbachevsky Ternopil National Medical University, 46001 Ternopil, Ukraine; kamyshnyi_om@tdmu.edu.ua

**Keywords:** chronic generalized periodontitis, Selenase, nitrosative stress, antioxidant system

## Abstract

Inflammatory periodontal diseases, despite all the efforts of modern dentistry, remain an important predictor of tooth loss worldwide. Oxidative stress plays a crucial role in the pathogenesis of periodontitis, making the use of antioxidants an attractive option for its treatment. Our attention was drawn to the selenium compound Selenase as an antioxidant therapeutic agent. In this study, we modeled a calcium-deficient prooxidant chronic generalized periodontitis (CGP) model in white non-linear rats. Then, after 14 days, Selenase (50 μg/kg) and Mexidol (250 mg/kg) were administered intragastrically. Blood samples from the animals were analyzed using ELISA and biochemical methods to determine Cu-Zn SOD, nitrotyrosine, GPX-4, iNOS, NO_x_, GSH, and GSSG levels. The CGP model led to the typical clinical signs of periodontitis, including hyperemia, edema, gingival pocket formation, bleeding, tooth mobility, as well as an increase in molecular–biochemical markers of nitrosative stress and a reduction of endogenous antioxidants in the blood. Selenase resulted in a decrease in the clinical manifestations of CGP, reduced iNOS, nitrotyrosine, and NO_x_ levels, and an increase in Cu-Zn SOD and GPX-4 compared to the control group (*p* < 0.05). Mexidol had a less pronounced effect on these markers compared to Selenase (*p* < 0.05).

## 1. Introduction

Numerous epidemiological studies have shown that chronic periodontitis is a major cause of tooth loss. Global estimates of the prevalence of severe periodontitis typically range from 10% to 15% [1]. Periodontal disease is characterized by a chronic bacterial infection of the oral cavity, which leads to gum inflammation and the gradual destruction of periodontal tissues and the alveolar bone that supports the teeth. The bacteria most likely to play an etiological role in the development of periodontal diseases include *Porphyromonas gingivalis*, *Actinobacillus actinomycetemcomitans*, *Tannerella forsythensis*, and *Treponema denticola* [2]. Pathogenic organisms initiate an inflammatory response in the surrounding tissues and begin to attack and destroy the alveolar bone and supporting structures around the teeth. The inflammatory response triggered by periodontal pathogens leads to an increase in proinflammatory cytokines—IL-1β, IL-6, TNF-α—and matrix metalloproteinases, which subsequently stimulate the production of reactive oxygen species (ROS) and induce oxidative stress [3]. Oxidative stress contributes to the damage of key cellular components, including DNA, proteins, and lipids [4,5].

Recently, it has been established that the interaction of phagocytic cells with invading pathogens leads to an increase in iNOS activity and elevated NO production, which, by reacting with the superoxide anion, forms cytotoxic species such as peroxynitrite. In our previous studies, we demonstrated that the modeling of periodontitis leads to an increase in NOS activity through the upregulation of iNOS expression and iNOS mRNA, accompanied by increased NO and nitrotyrosine levels, indicating the activation of nitrosative stress [6,7]. As a result of nitrosative stress, nitration of protein molecules and nucleic acids occurs, leading to a decrease in tissue reparative regeneration processes, immune suppression, disruption of molecular–biochemical mechanisms of cellular signaling, and enhanced apoptosis [8,9].

Antiseptics, antibiotics, astringents, and anti-inflammatory drugs are commonly used in the treatment of periodontitis [10,11]. Given the current understanding of the pathogenesis of periodontitis and the role of oxidative stress, antioxidants with various mechanisms of action are widely utilized—such as Thiotriazoline, α-tocopherol, recombinant superoxide dismutase preparations, propolis, mummy, and herbal complexes containing bioflavonoids. There is evidence supporting the use of these preparations in various well-known pharmaceutical forms and through different routes of administration, ranging from local to parenteral [12,13,14].

Considering the stages of oxidative stress formation and the mechanisms of action of modern antioxidants, the focus of pharmacologists and clinicians has shifted toward drugs that act as scavengers of reactive oxygen species (ROS) or modulators of antioxidant enzyme expression. In this context, selenium compounds have attracted attention, as they have proven to be excellent antioxidants, cardioprotectors, neuroprotectors, and apoptosis inhibitors. Research suggests that selenium derivatives can regulate ROS-dependent mechanisms of cellular signaling and endogenous cytoprotective mechanisms associated with heat shock proteins (HSP_70_) [15,16]. Our preliminary studies have shown promising results regarding the therapeutic effects of the selenium-based drug Selenase (sodium selenite) in the chronic periodontitis model in rats [12,17]. All of the above highlights the relevance and potential of this research.

The aim of the study: to evaluate the protective effect of the antioxidant Selenase in chronic generalized periodontitis by assessing its impact on the indicators of the antioxidant system and nitrosative stress in experimental animals.

## 2. Materials and Methods

### 2.1. Experimental Model

Experiments were conducted on 40 white non-linear rats, females, weighing 190–220 g, aged 8–9 weeks, obtained from the vivarium of the Institute of Pharmacology and Toxicology of the Ukrainian Academy of Medical Sciences. The duration of quarantine under the supervision of a qualified veterinarian (acclimatization period) for all animals was 14 days. During the quarantine, each animal was inspected daily (behavior and general condition), and the animals were observed twice a day in their cages (morbidity and mortality). Before the start of the study, animals meeting the inclusion criteria were randomly assigned to groups. The animals were placed in polycarbonate cages measuring 550 × 320 × 180 mm with galvanized steel lids measuring 660 × 370 × 140 mm and glass drinkers. Each cage housed no more than 5 rats. Each cage was equipped with a label indicating the study number, species, sex, animal numbers, and the medicinal preparations used. The room for animal housing maintained the following conditions: temperature—20–24 °C, humidity—30–70%, light cycle—12 h light/12 h dark. All rats were fed ad libitum with a standard diet for laboratory animals supplied by the company “Phoenix” (Zaporizhzhia, Ukraine). Water from the municipal water supply (after reverse osmosis and UV sterilization) was provided without restrictions. The bedding consisted of alder sawdust (*Alnus glutinosa*), which was pre-treated by autoclaving. The animals in the cage were marked with Diamond Green. All procedures were carried out according to the guidelines for the use of animals in biomedical experiments (Strasbourg, 1986, with amendments made in 1998) and the “European Convention for the Protection of Vertebrate Animals Used for Experimental and Scientific Purposes”. The experimental study protocols and their results were approved by the Ethics Committee of ZSMPhU.

The experimental model of chronic generalized periodontitis (CGP) was reproduced over 8 weeks using a calcium-deficient peroxide diet with reduced chewing function. The drinking water contained a 2% EDTA solution, and the animals were daily administered the prooxidant Delagil (chloroquine phosphate) at a dose of 30 mg/kg in the form of a 0.59% aqueous solution [18]. Throughout the entire experiment, the animals were fed soft food. After the formation of CGP, the animals received the experimental drugs intragastrically using a metal probe. All rats were divided into 4 groups (10 animals each):The intact group, animals that were planned to receive an intragastric administration of a 0.9% sodium chloride solution for 30 days;The control group, animals with experimental CGP that were given an intragastric administration of a 0.9% sodium chloride solution for 30 days;Animals with experimental CGP that received intragastric administration of the Selenase preparation, 50 μg/kg, through an atraumatic metal probe for 30 days [19];Animals with experimental CGP that received daily intragastric administration of the reference drug Mexidol, 250 mg/kg, for 30 days [18].

The study used Selenase (sodium selenite pentahydrate) (Mivolis, Karlsruhe, Germany) and Mexidol (2-Ethyl-6-methyl-3-hydroxypyridine succinate) (Meksikor, CJSC “Tekhnolog”, Uman, Ukraine).

### 2.2. Anesthesia

Under the administration of thiopental anesthesia (40 mg/kg), rats from all experimental groups were taken out of the study. Following this, blood samples were obtained from the celiac artery for subsequent analysis.

### 2.3. Preparation of Biological Material

Blood was taken from the abdominal aorta by syringe, and serum was separated by centrifugation at +4 C at 1500 rpm for 20 min [20] on an Eppendorf 5804R centrifuge.

### 2.4. Enzyme-Linked Immunosorbent Assay (ELISA)

Nitrotyrosine was assessed in the blood serum using the solid-phase immunoassay sandwich method of ELISA. The ELISA Kit (Catalog No. HK 501-02) from Hycult Biotech, Uden, The Netherlands was used according to the instructions.

The activity of inducible nitric oxide synthase (iNOS) in the blood serum was assessed through enzyme immunoassay using the MyBioSource kit (MyBioSource, San Diego, CA, USA, #MBS023874), according to the instructions.

The activity of Cu-Zn-dependent superoxide dismutase was assessed in the blood serum using the Rat SOD1/Cu-Zn SOD (Sandwich ELISA) ELISA Kit—LS-F4234 from LSbio (Lynnwood, WA, USA) according to the instructions.

The activity of glutathione peroxidase-4 was assessed in the blood serum using the GPX-4 (Rat Glutathione Peroxidase 4, Phospholipid hydroperoxide glutathione) ELISA Kit from MyBioSource, Inc. (San Diego, CA, USA). Catalog #MBS934198 was used according to the instructions.

These analyses were conducted on a complete plate enzyme immunoassay analyzer (SIRIO-S, Seac, Bologna, Italy).

### 2.5. Biochemical Methods

The levels of NO metabolites (NO_x_) in the hearts were assessed using the Griess method. The serum obtained as above (1.0 mL) was deproteinized by adding 100 µL of 0.092 M zinc sulfate and 100 µL of 1 M NaOH, stirred, and left for 30–40 min. Then, it was centrifuged at 4000× *g* for 10 min (at 5 C) using an Eppendorf™ 5430 G centrifuge (Hamburg, Germany). Subsequently, 100 µL of the resulting supernatant was transferred to a well in a microplate, and 0.5 mM of vanadium (III) chloride was added to each well to reduce nitrate to nitrite. Following this, 50 µm of sulfonamide and 0.2 µm of N-1-(naphthyl) ethylenediamine were added. The total volume of the incubation mixture is 300 µL. The next step was to incubate the samples for 30 min at 37 °C, and the optical density was measured at 540 nm. The concentration of NO_x_ was determined using a linear standard curve within the range of 0–50 µµmol/L sodium nitrate. NO_x_ levels in the tissues were expressed in µmol/L. Measurements were performed using the Eppendorf BioSpectrometer spectrophotometer (Eppendorf, Hampton, NH, USA). The determination of reduced and oxidized glutathione was carried out fluorometrically. The method is based on the interaction of ortho-phthalic anhydride with reduced glutathione, resulting in the formation of a fluorescent complex, which is detected fluorometrically at Ex/Em = 340/420 nm. In a test tube containing 2.0 mL of 0.5 M phosphate buffer (pH 8.0), 0.1 mL of blood serum and 0.5 mL of 1% ortho-phthalic anhydride are added. The reaction mixture is mixed and incubated for 5 min, after which it is fluorometrically measured at Ex/Em = 340/420 nm.

To determine oxidized glutathione, in a test tube containing 2.0 mL of 0.5 M phosphate buffer (pH 12.0), 0.1 mL of blood serum is added. To mask the reduced glutathione, 0.04 mL of 0.5 mM 1-methyl-4-vinyl-pyridine is added. The reaction mixture is incubated for 60 min at room temperature. To reduce the oxidized glutathione, 0.1 mL of the reducing mixture (glutathione reductase 38 units, 7 mg NADPH dissolved in 20 mL of 0.5 M phosphate buffer pH 7.4) is added to the sample. Reduction is carried out for 2 min at 37 °C. Then, 0.5 mL of ortho-phthalic anhydride is added, and it is fluorometrically measured at Ex/Em = 340/420 nm. Measurements were performed on a fluorescence spectrophotometer (Agilent Fluorescence-spectrophotometer, Santa Clara, CA, USA).

### 2.6. Statistical Analysis

Experimental data were statistically analyzed using “StatisticaR for Windows 6.0” (StatSoft Inc., № AXXR712D833214FAN5, Kraków, Poland), “SPSS16.0”, and “Microsoft Office Excel 2010” software. Prior to statistical tests, we checked the results for normality (Shapiro–Wilk and Kolmogorov–Smirnov tests). In a normal distribution, intergroup differences were considered statistically significant based on the parametric Student’s *t*-test. If the distribution was not normal the comparative analysis was conducted using the non-parametric Mann–Whitney U test. To compare independent variables in more than two selections, we applied ANOVA dispersion analysis for a normal distribution and the Kruskal–Wallis test for a non-normal distribution. To analyze correlations between parameters, we used correlation analysis based on the Pearson or Spearman correlation coefficient. For all types of analysis, the differences were considered statistically significant at *p* < 0.05 (95%).

## 3. Results

Our research has shown that the modeling of chronic generalized periodontitis (CGP) led to typical manifestations of the disease, including bleeding, hyperemia, gum swelling, and tooth mobility. The depth of the periodontal pocket was 8 mm. In rats with experimental CGP receiving a therapeutic course of Selenase at a single dose of 50 μg/kg, a pronounced therapeutic effect was observed—significant reduction of the periodontal pocket size to 4.6 mm, along with a substantial decrease in bleeding and swelling. In contrast, the course administration of Mexidol at a single dose of 250 mg/kg intragastrically to rats with CGP resulted in a less pronounced therapeutic effect compared to the group receiving Selenase. In this group of animals, gum swelling persisted, though it was smaller than in the control group. Bleeding upon probing of the periodontal pocket was still present, the depth of the periodontal pocket was 6 mm, and tooth mobility remained unchanged.

Molecular and biochemical blood tests of the control group rats (CGP without treatment) revealed a significant decrease in the concentration of antioxidant enzymes—glutathione peroxidase-4 (GPX-4) by 65.5% and Cu/Zn-dependent superoxide dismutase (Cu/Zn SOD) by 37.4%, along with a decrease in the concentration of reduced glutathione by 52.5% and an increase in its oxidized form by 132% (*p* < 0.05). These disturbances in the antioxidant system of the blood in CGP animals occurred against the backdrop of oxidative stress activation—an increase in nitrotyrosine levels by 4.3 times, iNOS by 2.3 times, and NO metabolites by 78.4% (*p* < 0.05). Thus, the modeling of CGP using this method led to significant activation of oxidative stress.

The course administration of Selenase at a single dose of 50 µg/kg intragastrically to rats with chronic generalized periodontitis (CGP) had a significant effect on the molecular and biochemical parameters of the antioxidant system and nitrosative stress (Table 1 and Table 2). In the blood of animals in this group, the concentration of nitrotyrosine was 43.1% lower than in the control group (*p* < 0.05). Additionally, the levels of iNOS and NO_x_ decreased by 16.6% and 26.8%, respectively (*p* < 0.05).

The administration of Selenase to rats with CGP also had a positive effect on the expression of antioxidant enzymes and the thiol-disulfide system in the blood’s glutathione pool. Specifically, the Cu/Zn SOD levels increased by 40.4% compared to the control group (*p* < 0.05). Selenase demonstrated a pronounced stimulating effect on the glutathione component of the thiol-disulfide system in rats with CGP, with GPX-4 levels in the serum increasing 2.2 times compared to the control group (*p* < 0.05). Furthermore, the level of reduced glutathione increased by 90% (*p* < 0.05), while the concentration of its oxidized form decreased by 53.5% in the blood of CGP rats (*p* < 0.05).

The administration of Mexidol to rats with CGP as a medicinal agent had a significant effect only on the nitrotyrosine level, which is consistent with previous studies [16]. Mexidol also increased the concentration of reduced glutathione in the blood of rats with CGP (*p* < 0.05). However, in terms of its impact on the markers of nitrosative stress and the antioxidant system, Mexidol was less effective than Selenase in influencing parameters such as GPX-4, Cu/Zn SOD, nitrotyrosine, GSH, and GSSG in the blood of CGP rats (*p* < 0.05) (Table 1 and Table 2).

The obtained results characterize Selenase as a highly effective antioxidant agent.

## 4. Discussion

The results we obtained regarding the activation of oxidative stress reactions in chronic periodontitis align with data from other researchers and clinicians, confirming their consistency. The onset and progression of periodontitis are closely linked to pathogenic microorganisms. Reactive oxygen species (ROS), such as superoxide, hydrogen peroxide, and hydroxyl radicals—products of normal cellular metabolism—play a role in combating these bacterial pathogens [12,21]. The production of ROS is a crucial protective mechanism against diseases associated with phagocytic infiltration, acting as a defense against bacterial infections. Simultaneously, in response to bacterial colonization, immune system leukocytes release proinflammatory cytokines that are central to the progression of chronic periodontitis, including IL-1α, IL-1β, IL-6, IL-12, tumor necrosis factor (TNF)-α, as well as regulatory cytokines like IL-4, IL-1 receptor antagonist (RA), IL-10, and induced protein IP-10 [22,23,24]. IL-1β at the site of inflammation contributes to increased local blood flow, leukocyte recruitment, and neutrophil infiltration. It also activates the expression of MMP-9, initiates the expression of mRNA for iNOS, and enhances ROS production. Another key pathway of ROS formation in periodontitis is the NADPH oxidase reaction in neutrophils. It has been shown that NADPH oxidase activity correlates with IL-1β expression in periodontal inflammatory diseases [25,26]. The hyperproduction of ROS results in oxidative modification of macromolecules (proteins, nucleic acids, and lipids), leading to tissue damage (Figure 1).

Clinical studies have shown that periodontitis is correlated with an increase in stable products of lipid peroxidation in saliva and gingival crevicular fluid. Experimental studies have also demonstrated that modeling periodontitis leads to an increase in the blood levels of oxidative modification products of nucleic acids, prostaglandins, fatty acids, proteins, as well as hydrogen peroxide in polymorphonuclear leukocytes [13,27]. As periodontitis progresses, ROS produced by periodontal inflammation diffuse into the bloodstream, leading to systemic oxidative stress, which can gradually affect multiple organs. Thus, the oxidative stress induced by periodontitis can have a negative impact on overall health.

Cytotoxic forms of nitric oxide (NO), which are involved in alveolar bone resorption, also play a role in initiating oxidative stress in periodontitis [28,29,30]. The increased expression of iNOS, along with the elevated concentration of NO in the blood of rats with CGP, as demonstrated in our study, is consistent with previous research. NO plays a crucial role in inflammatory processes, and the inducible form, iNOS, is involved in inflammatory reactions in periodontitis. Phagocytic cell interactions with invading pathogens lead to NO formation, accompanied by increased iNOS expression and ROS production, which results in the formation of peroxynitrite responsible for the destruction of microorganisms. Products of bacterial cell wall degradation and peroxynitrite trigger a positive feedback loop, further inducing iNOS and excessive production of NO and its derivatives.

It is important to emphasize that excessive production of NO derivatives can lead to cytotoxicity against the host’s own tissues, causing damage through oxidation and nitration reactions, inhibition of mitochondrial enzymes, and DNA damage. Tissue affected by periodontitis shows a higher level of iNOS expression compared to healthy tissue, with macrophages and endothelial cells being the sources of iNOS in periodontal tissues. The level of NO in the saliva of patients with gingivitis and periodontitis was higher than in the healthy control group. Moreover, as the disease progresses to a more destructive stage, the inflammatory process intensifies, leading to an increase in NO levels. This allows for the use of stable NO metabolite levels in saliva as a determining factor of the inflammatory status in patients [31,32,33,34].

Increased NO production in inflammatory periodontal diseases leads to the formation of nitrosative stress, as evidenced by elevated nitrotyrosine levels in blood or tissues. Nitrosative stress may play a crucial role in exacerbating periodontitis in patients [6,31]. As a result of nitrosative stress in periodontitis, the apoptosis of periodontal ligament cells is initiated, localized microvasculopathies are formed, local ischemia of the periodontium occurs, and there is dysfunction of the endothelium of the periodontal microvessels [8,9]. The reactivity of nitric oxide (NO) forms triggers a series of harmful reactions, including lipid peroxidation, protein oxidation, and DNA damage. Alveolar bone resorption, connective tissue degradation, and periodontium inflammation are additional conditions exacerbated by these processes (Figure 2). Furthermore, the delicate balance between antioxidants, particularly intermediates of the thiol-disulfide system, and cytotoxic forms of NO is disrupted by oxidative stress, which impairs antioxidant defense systems and exacerbates tissue damage in the periodontium [29,35,36].

Suppression of Zn/Cu SOD expression in inflammatory periodontal diseases increases the production of inflammasome components of the NOD-like receptor protein 3 (NLRP3) inflammasome. Given that Zn/Cu SOD exerts a protective effect by inhibiting the NLRP–caspase-1–IL-1β inflammasome axis during inflammation, the reduction in expression and activity of this enzyme not only intensifies inflammation but also leads to a recurrent surge in ROS production during the course of inflammatory reactions in the periodontium [37]. The observed decrease in plasma levels of CGP, GPX-4 expression, and the increase in GSSG in rats have been confirmed by other studies and may be associated with both the activation of free radical reactions and the deprivation of specific components of the glutathione system. In patients with periodontitis, there is a reduction in the overall antioxidant capacity in saliva, as well as lower concentrations of reduced glutathione (GSH) in serum and gingival fluid. Periodontal therapy restores the redox balance. Therapeutic considerations regarding the supplementary use of glutathione or pharmacological agents that modulate its cellular levels in the treatment of periodontitis, to limit tissue damage related to oxidative stress and improve wound healing, should not be underestimated [38]. It has been shown that pharmacological elevation of GSH levels in monocytes and macrophages blocks H_2_O_2_-dependent activation of NF-κB and, indirectly, the proinflammatory cytokines TNF-α and IL-1β. An excess of cytotoxic forms of NO leads to the nitration of proteins involved in GSH synthesis, resulting in a decrease in its concentration [9,39]. GPX-4 is the main regulator of ferroptosis, interrupting lipid peroxidation by converting lipid hydroperoxides into non-toxic lipid alcohols. Therefore, the reduction in its expression in the plasma of rats with CGP is interpreted as an enhancement of lipid peroxidation processes, particularly of arachidonic acid, leading to a surge in inflammation through the increased production of proinflammatory prostaglandins and stable carbonylated products (ketones, aldehydes, trans-2-nonenal). Additionally, the inhibition of GPX-4 triggers ferroptosis and the death of osteocytic cells [40]. It is known that increased production of TNF-α, IL-1β, IL-6, and IL-8 occurs in connection with GSH deficiency, and its modulation not only reduces oxidative stress activity but also inflammation in the periodontium [41,42]. There is an opinion that the glutathione component of the thiol-disulfide system is stimulated by selenium derivatives. The protective action of selenium in mammalian cells is mediated by seleno-amino acids, either selenocysteine or selenomethionine. The active site of powerful glutathione peroxidases (GPXs) contains selenocysteine residues. Additionally, it has been shown that other selenoproteins (such as selenoprotein P and thioredoxin reductase) also possess antioxidant properties [43,44]. Selenium may play a role in the regulation of redox states of proteins, such as NF-κB transcription factors, by regulating the synthesis of enzymes (antioxidants, those involved in energy metabolism, and glutathione synthesis) (Figure 3).

An important link in the cytoprotective action of selenium derivatives is the activation of GSH-dependent mechanisms, which is implemented through the increased expression of HSP_70_ [15,45]. The role of selenium as a cytoprotector is linked both to its antioxidant properties and its ability to prevent inflammation, autophagy, as well as internal and external apoptosis pathways. Signaling pathways such as p-AMPK, PARP, NRF2, and STAT are involved in the protective effects of selenium [17,46]. The most significant effect of Selenase has been demonstrated through its ability to positively influence glutathione synthesis, modulate the expression of seleno-glutathione peroxidase, and affect GSH-dependent mechanisms of endogenous cyto- and neuroprotection [47,48]. Mexidol can reduce the production of proinflammatory products, such as arachidonic acid metabolites, aldehydes, trans-2-nonenals, and lipid peroxides, due to its properties as a direct antioxidant and an inhibitor of lipoperoxidation processes. These properties may provide certain therapeutic effects in chronic liver disease (CLD). Additionally, it is possible that Mexidol can interrupt ROS-dependent mechanisms of IL-1β expression [7,18].

## 5. Limitations of the Study

All experimental animals were housed in groups of five per cage. During the modeling of chronic generalized periodontitis (CGP) and the oral cavity examination, the experimental animals experienced additional stress. This CGP model does not fully replicate the clinical picture of chronic generalized periodontitis, as it does not simulate the complete combination of pathogenic factors (such as bacterial colonization, mechanical damage, microcirculatory disturbances, etc.) affecting periodontal tissues in laboratory animals. Additionally, rats have an inherent resistance to periodontal diseases and a microbial composition that differs from that of humans. The study did not investigate the long-term therapeutic effects of the drugs or the preservation of their efficacy over extended periods.

## 6. Perspectives for Further Research

The obtained results provide experimental evidence supporting the need for further in-depth studies on Selenase, with the goal of justifying its use in the complex therapy of chronic generalized periodontitis. The development and creation of a new dosage form of Selenase—specifically a dental gel—are also considered, along with further research into its effectiveness and safety.

## 7. Conclusions

1. The calcium-deficient prooxidant model of chronic generalized periodontitis (CGP) leads to a typical clinical picture of periodontitis—hyperemia, edema, the formation of periodontal pockets, bleeding, tooth mobility, and an increase in molecular–biochemical markers of nitrosative stress in the blood, such as iNOS, NO_x_, and nitrotyrosine, accompanied by a decrease in glutathione levels and the expression of glutathione peroxidase-4 and Zn/Cu-dependent superoxide dismutase.

2. The course administration of the selenium preparation Selenase (50 µg/kg) to animals with CGP as part of the therapeutic regimen led to a reduction in the periodontal pocket to 4.6 mm and almost complete absence of bleeding and edema.

3. The administration of Selenase resulted in the inhibition of nitrosative stress in rats with CGP, as evidenced by a decrease (*p* < 0.05) in the expression of iNOS and lower concentrations of nitrotyrosine and NO_x_ in the blood of the animals.

4. The manifestation of the antioxidant action of Selenase in CGP was an increase (*p* < 0.05) in the concentration of reduced glutathione (GSH), the expression of glutathione peroxidase-4 (GPX-4), and Cu/Zn superoxide dismutase (SOD) in the blood of the animals.

5. Selenase surpassed (*p* < 0.05) the effectiveness of Mexidol in influencing such indicators as GPX, Cu/Zn SOD, GSH, nitrotyrosine, and iNOS.

## Figures and Tables

**Figure 1 cimb-47-00186-f001:**
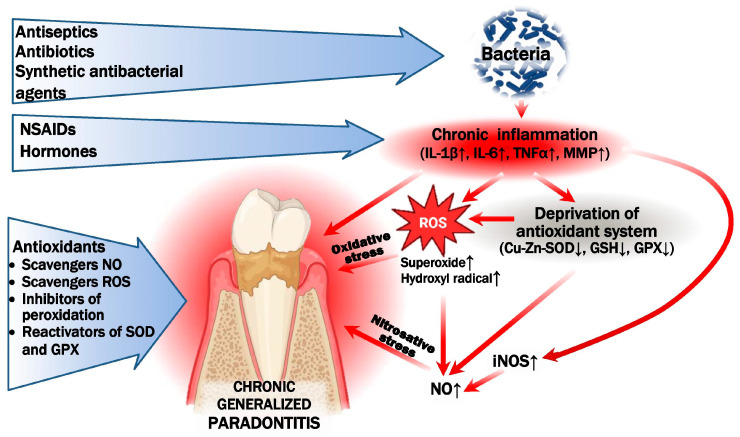
The main molecular and biochemical links in the pathogenesis of chronic periodontitis and the primary methods of pharmacocorrection. Periodontal disease is characterized by a chronic bacterial infection of the oral cavity (*Porphyromonas gingivalis*, *Actinobacillus actinomycetemcomitans*, *Tannerella forsythensis*, and *Treponema denticola*), which leads to inflammation of the gums (increased IL-1β, IL-6, TNF-α, and matrix metalloproteinases), activation of reactive oxygen species (ROS), and deprivation of antioxidant systems, resulting in the gradual destruction of periodontal tissues and alveolar bone supporting the teeth. Treatment for periodontitis includes antiseptics, antibiotics, astringents, anti-inflammatory drugs and considering the role of oxidative stress and antioxidants. ↑ means increase, ↓ means decrease.

**Figure 2 cimb-47-00186-f002:**
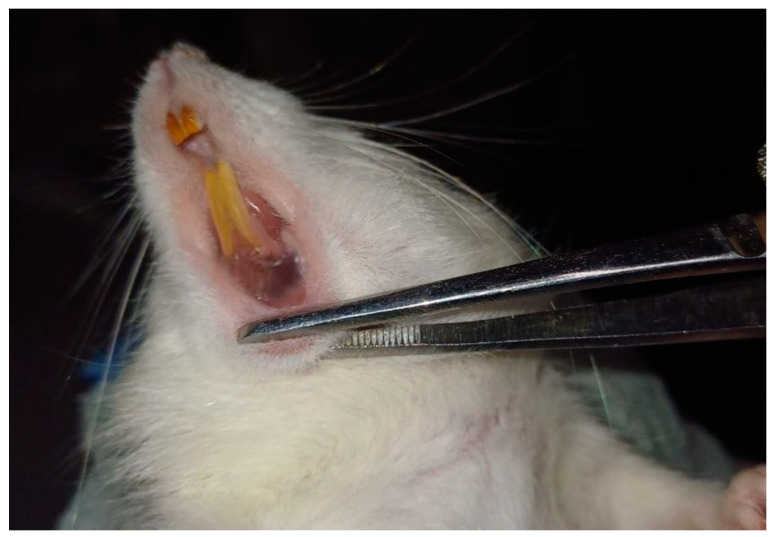
Rat of control group (CGP without treatment) on the 45th day of observation—bleeding, hyperemia, gingival edema, tooth mobility, and deep gingival pockets were recorded.

**Figure 3 cimb-47-00186-f003:**
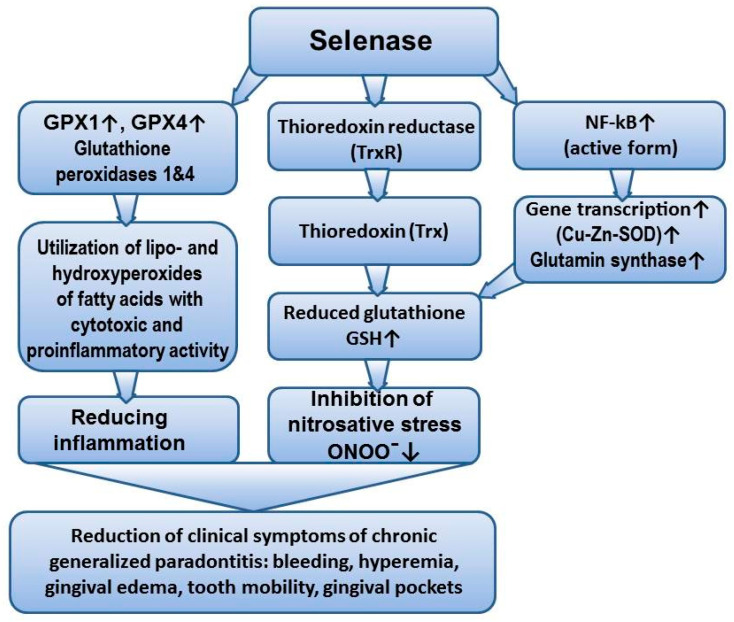
The mechanism of action of selenium derivative—Selenase. Selenase stimulates the glutathione link of the thiol-disulfide system, particularly enhancing the expression and activity of powerful glutathione peroxidases (GPXs). Selenase regulates the synthesis of enzymes (antioxidants, participants in energy metabolism, and glutathione synthesis) by influencing the activity of the transcription factor NF-κB. Selenase activates GSH-dependent mechanisms of endogenous cytoprotection, realized through increased expression of HSP_70_. Selenium prevents inflammation, autophagy, as well as both internal and external pathways of apoptosis. ↑ means increase, ↓ means decrease.

**Table 1 cimb-47-00186-t001:** Molecular markers of nitrosative stress and antioxidant system in the blood of animals with experimental CGP and under the influence of pharmacological treatment.

Indicators	Intact (*n* = 10)	CGP, Control (*n* = 10)	CGP + Selenase (50 µg/kg) (*n* = 10)	CGP + Mexidol (250 mg/kg) (*n* = 10)
Periodontal pocket depth, mm	0	8.0 ± 0.43 ^1^	4.6 ± 0.69 ^1^*	6.0 ± 0.93 ^1^*
Nitrotyrosine, ng/mL	50.5 ± 3.7	217.7 ± 15.21	123.7 ± 10.8 *^1#^	167.5 ± 9.7 *^1^
iNOS, ng/mL	32.7 ± 2.55	76.4 ± 5.12 ^1^	63.7 ± 6.12 *^1^	72.3 ± 5.45 ^1^
GPX-4, pg/mL	48.7 ± 2.33	17.7 ± 1.28 ^1^	39.2 ± 4.12 *^1#^	21.8 ± 2.02 ^1^
Cu/Zn SOD, pg/mL	88.5 ± 7.44	55.4 ± 3.40 ^1^	77.8 ± 5.25 *^1#^	62.8 ± 4.52 ^1^

*—changes are significant in relation to animals in the control (CGP) group (*p* < 0.05); ^1^—changes are significant in relation to animals of the intact group (*p* < 0.05); ^#^—changes are significant in relation to the group of animals with CGP that received Mexidol (*p* < 0.05).

**Table 2 cimb-47-00186-t002:** Biochemical markers of nitrosative stress and the antioxidant system in the blood of animals with experimental CGP and under the influence of pharmacological agents.

Indicators	Intact (*n* = 10)	CGP, Control (*n* = 10)	CGP + Selenase (50 µg/kg) (*n* = 10)	CGP + Mexidol (250 mg/kg) (*n* = 10)
NO metabolites (NO_x_), µmol/L	6.5 ± 4.7	11.2 ± 1.2 ^1^	8.2 ± 0.5 *^1^	6.50 ± 4.7
GSH, µmol/L	678.5 ± 45.0	321.8 ± 21.2 ^1^	611.5 ± 27.2 *^#^	419.4 ± 21.4 *
GSSG, µmol/L	38.1 ± 2.8	88.5 ± 6.3 ^1^	41.1 ± 3.7 *^#^	77.1 ± 5.2 ^1^

*—changes are significant in relation to animals in the control (CGP) group (*p* < 0.05); ^1^—changes are significant in relation to animals of the intact group (*p* < 0.05); ^#^—changes are significant in relation to the group of animals with CGP that received Mexidol (*p* < 0.05).

## Data Availability

Data is contained within the article.

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
