# Peer review of "Antioxidant Mechanisms of the Protective Action of Selenase in Experimental Chronic Generalized Periodontitis"

_cimb, 2025, doi:10.3390/cimb47030186_

Round 1
Reviewer 1 Report
Comments and Suggestions for Authors
Overall, the main issue with this manuscript is that it does not clearly explain the relationship between inflammation, oxidative stress, nitrosative stress, and the development of CGP, nor how antioxidant treatment addresses these mechanisms. While these concepts are mentioned separately, their interactions and contributions to CGP pathophysiology are not well integrated. I strongly recommend adding a clear summary in the manuscript.
- In line 71, the study's aim is stated as: "to assess the effect of Selenase on the indicators of nitrosative stress and the antioxidant system in rats with experimental CGP." However, the Introduction section does not mention nitrosative stress, only talking the relationship between CGP, inflammation, and oxidative stress. I suggest:
-Including a discussion on nitrosative stress in the Introduction, explaining its role in CGP and how it differs from oxidative stress.
-Rewording the study objective to ensure consistency with the background provided.
2. Figure 1 is visually confusing and lacks clear differentiation between the treatment approaches for CGP (on the left) and the causative factors of CGP (on the right). The absence of clear boundaries between these two categories makes interpretation difficult. Additionally, there is overlap among the listed causative factors. For example, insufficient endogenous antioxidants is more likely a consequence of inflammation rather than an independent cause. This misclassification may lead to misunderstandings regarding the pathophysiological mechanisms of CGP.
3. Figure 3 illustrates the mechanism of Selenase in treating CGP; however, it does not establish a clear connection to oxidative stress, nitrosative stress, and inflammation. This lack of integration makes it difficult for readers to fully understand how Selenase mechanistically interacts with the pathophysiology of CGP.
Author Response
We thank the reviewer for evaluating our work and valuable feedback.
We have now rephrased the abstract to increase its clarity.
We have also supplemented the "Introduction" section and revised the aims of the study.
Additionally, we have improved the graphic design of Figures 1 and 3.
Reviewer 2 Report
Comments and Suggestions for Authors
The authors in this manuscript entitled ,, Antioxidant Mechanisms of the Protective Action of Selenase in Experimental Chronic Generalized Periodontitis” evaluated the effect of a selenium-based preparation on molecular and biochemical markers of oxidative stress in an experimental study on rats with periodontitis.
The manuscript has some major issues that need to be corrected and some unclear question, to which I ask the authors to answer:
The animals used in experiment and the environmental condition must be reported according to ARRIVE guidelines (Percie du Sert N, Ahluwalia A, Alam S, Avey MT, Baker M, et al. (2020) Reporting animal research: Explanation and elaboration for the ARRIVE guidelines 2.0. PLOS Biology 18(7): e3000411. https://doi.org/10.1371/journal.pbio.3000411 ). The manuscript does not mention the sex, age of the animals, strain, housing conditions, microclimate factors, etc.
How did you determine the number of animals for experimentation? Did you use any statistical software?
What is the logic behind using a group of animals that you did not perform any procedure on, except the final one? Using such a group contradicts the 3R principles.
How long did the induction of periodontitis last and how long after induction did the treatment begin? There are inconsistencies between the data in the abstract and the experimental model chapter and confusion may arise.
How did you determine that chronic periodontitis had set in?
For group 3 - was Selenase administered daily? – line 98
The anesthesia chapter is confusing. thiopental provides light anesthesia at a dose of 40 mg/kg. Was blood collected from the celiac artery under this anesthesia?
Was the blood collected from both the celiac artery and the aorta according to the manuscript? when? were there intermediate blood collections or just one at the end?
Was the collection from the aorta and celiac artery done with the abdominal cavity opened or not?
How did you euthanize the animals?
In the results chapter, I recommend making graphs with the statistical expression of the results, not just tables and text. Otherwise, it is difficult to highlight the difference between the groups.
Line 196 - chronic gastric pathology (CGP) - what gastric pathology is about?
To write the software program with which figures 1 and 3 were made.
In the discussion chapter, a few paragraphs should be written about the limitations of the study and possible future research
The conclusions should not repeat the results.
Comments on the Quality of English LanguageThe English in this manuscript needs improvement. Consult a native speaker or editing software.
Author Response
We thank the reviewer for valuable suggestions and feedback overall.
In our study involving animals, their selection, housing, and usage were conducted following both international and national guidelines. For the preclinical evaluation of the pharmacological action of the potential drug, in accordance with the State Expert Center of the Ministry of Health of Ukraine and international recommendations, we used an adequate number of animals (10 rats per group) to obtain reliable results. Additional information about the animals has been included in the "Materials and Methods" section. Statistical methods are specified in section 2.6.
For the intact control group, which did not undergo any procedures, except for the final one, we administered physiological saline through a metallic probe into the stomach in acceptable volumes, similar to the volumes administered to the other experimental groups. Experimental periodontitis was assessed by changes in gum color and consistency, presence of gum bleeding upon probing, tooth mobility, probing depth of the gingival sulcus and periodontal pocket, as well as an increase in inflammation markers – C-reactive protein, IL-1b, and TNF-a. In our previous studies, the reproducibility of this periodontitis model has been confirmed histologically by signs of inflammation in the periodontal tissues.
Thiopental at a dose of 40 mg/kg provides good anesthesia if fresh, allowing for animal examination and blood collection from the abdominal aorta. Doses exceeding 50-60 mg/kg of thiopental often lead to premature death of some animals, based on our experience. Selenase and Mexidol were administered daily at the same time. Blood samples were only taken at the end of the experiment from the abdominal aorta, before bifurcation (e.g., Anatomy of the Rat by Nоzdrachev A.D., Polyakov E.L., St. Petersburg, 2001). After blood and periodontal tissue collection, rats were euthanized using an overdose of sodium thiopental.
The authors would like to consider the option to present the data in a table format. Absolute values are more objective and provide the opportunity to assess the accuracy of biochemical and molecular marker determination methods. We have now made the necessary additions to the manuscript.
Round 2
Reviewer 1 Report
Comments and Suggestions for Authors
I appreciate the revisions made to the manuscript. The updated Figure 1 and Figure 3 are now clearer and more informative. I am generally satisfied with this version of the manuscript. However, I noticed that the semicolons in the figures should be replaced with commas. Please make this minor correction.
Author Response
We thank the reviewer for this comment.
The Figures are now corrected as suggested.
Reviewer 2 Report
Comments and Suggestions for Authors
The authors provided some explanations and clarifications to my observations.
They also made additions and corrections in the text, but some of them were not marked in green in second manuscript nor detailed in the explanations.
However, there are other observations:
throughout the text the names of the bacteria should be written in italics
a few paragraphs about the limitations of the study should be added
Line 151 - a word is in Ukrainian
Author Response
We thank the reviewer for these additional comments.
Previously, we have detailed our changes in the response and incorporated the suggestions for improving the paper.
The text highlighted in green as follows:
in section 2 "Introduction," information explaining the relationship between inflammation, oxidative stress, and nitrosative stress in the development of CGP (Reviewer 1's request) has been added; corrections to the research aim (Reviewer 1's request) have been made;
in the "Materials and Methods" section, information about the animals (your request) has been added, and
in the "Discussion" section, information about the limitations of the study and the prospects of possible future research has been included.
Moreover, the whole text has been revised to improve English grammar and style, including line 151, so there may be minor changes throughout the text. The changes in the text preserved its content and meaning.
Round 3
Reviewer 2 Report
Comments and Suggestions for Authors
The Authors have addressed all the remarks. The article can be published.